# Enantioselective functionalization of unactivated C(sp³)–H bonds through copper-catalyzed diyne cyclization by kinetic resolution

Yang-Bo Chen[1], Li-Gao Liu[1], Zhe-Qi Wang[1], Rong Chang[1], Xin Lu [1]✉, Bo Zhou [1] & Long-Wu Ye [1,2]✉

Site- and stereoselective C–H functionalization is highly challenging in the synthetic chemistry community. Although the chemistry of vinyl cations has been vigorously studied in C(sp³)–H functionalization reactions, the catalytic enantioselective C(sp³)–H functionalization based on vinyl cations, especially for an unactivated C(sp³)–H bond, has scarcely explored. Here, we report an asymmetric copper-catalyzed tandem diyne cyclization/unactivated C(sp³)–H insertion reaction via a kinetic resolution, affording both chiral polycyclic pyrroles and diynes with generally excellent enantioselectivities and excellent selectivity factors (up to 750). Importantly, this reaction demonstrates a metal-catalyzed enantioselective unactivated C(sp³)–H functionalization via vinyl cation and constitutes a kinetic resolution reaction based on diyne cyclization. Theoretical calculations further support the mechanism of vinyl cation-involved C(sp³)–H insertion reaction and elucidate the origin of enantioselectivity.

Carbon-hydrogen bond is one of the most common chemical bonds in organic frameworks. The enantioselective functionalization of C–H bonds is recognized as a very attractive strategy for synthesizing complex chiral molecules and has been widely used in organic synthesis and pharmaceutical chemistry[1–6]. In the past decades, several different approaches have been developed to modify C(sp³)–H bonds enantioselectively[7,8], including transition metal-catalyzed C–H activation[9–11], C–H insertion to a metal carbene or nitrene[12–15], and biomimetic catalysis via radical reaction[16–18]. However, these C–H functionalizations generally involved a relatively activated C(sp³)–H bond next to a heteroatom or at a benzylic and allylic position. In contrast, unactivated C(sp³)–H functionalization is rarely reported, as these C–H bonds possess high bond energy and a similar chemical environment, resulting in site- and stereoselectivity challenges.

Catalytic kinetic resolution (KR) is a powerful and practical method to produce enantioenriched compounds[19–21], and has been widely utilized in academia and industry in the past decades, especially for the catalytic asymmetric synthesis of chiral alcohols, amines, etc.[22–27]. However, the kinetic resolution based on alkynes has rarely been reported. In 2005, Fokin et al. reported the first enantioselective azide-alkyne cycloaddition reaction through kinetic resolution albeit with low selectivity factors[28]. After that, significant progress based on the azide-alkyne cycloaddition reaction was achieved by Zhou et al.[29–33], which provided a series of chiral α-tertiary 1,2,3-triazoles (Fig. 1a). In addition to this strategy, a few examples of kinetic resolution reactions based on alkynes via cyclization have also been invoked[34–40], but the nucleophiles are limited to alcohols[34,35], alkenes[36,37], and acyl groups[38–40] (Fig. 1b). To the best of our knowledge,

[1]State Key Laboratory of Physical Chemistry of Solid Surfaces, Key Laboratory of Chemical Biology of Fujian Province and College of Chemistry and Chemical Engineering, Xiamen University, Xiamen 361005, China. [2]State Key Laboratory of Organometallic Chemistry, Shanghai Institute of Organic Chemistry, Chinese Academy of Sciences, Shanghai 200032, China. ✉e-mail: xinlu@xmu.edu.cn; longwuye@xmu.edu.cn

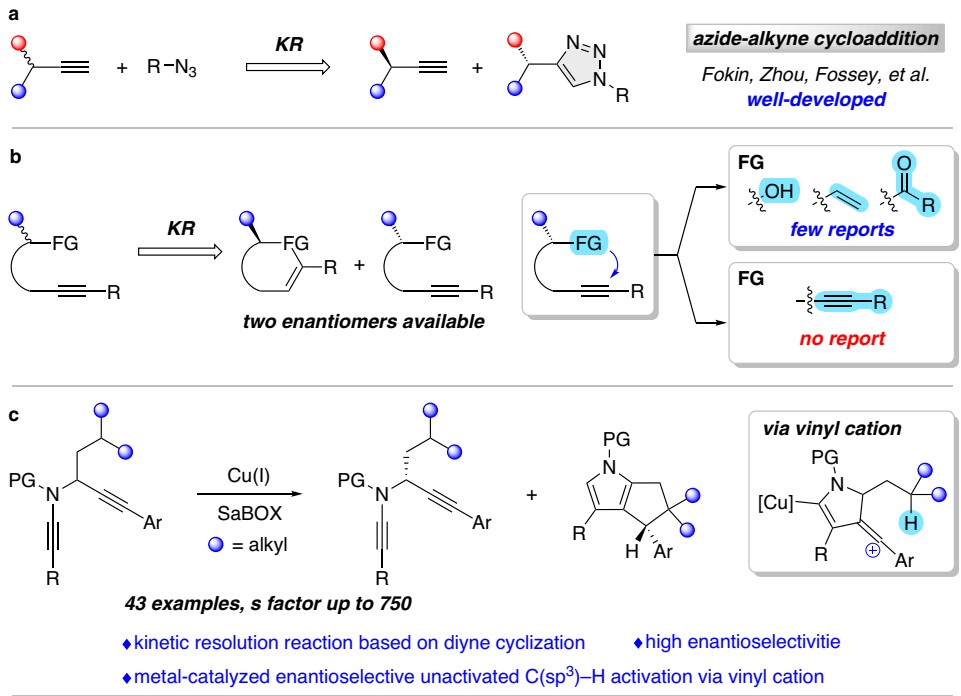

**Fig. 1 | Kinetic resolution reactions for asymmetric alkyne transformation.**
**a** Kinetic resolution reactions based on azide-alkyne cycloaddition. **b** Kinetic resolution reactions based on alkynyl cyclization. **c** This work: enantioselective functionalization of unactivated C(sp³)−H bonds through copper-catalyzed diyne cyclization via kinetic resolution. KR kinetic resolution, FG functional group, PG protecting group.

the kinetic resolution of alkynes via direct diyne cyclization[41–44] has not yet been realized.

Over the past decades, the chemistry of vinyl cations has attracted extensive attention due to their unique carbene-like reactivity[45,46], and has been well explored in unactivated C(sp³)−H functionalization reactions[47–53]. However, the catalytic enantioselective C(sp³)−H functionalization based on vinyl cations is extremely difficult, likely due to the lack of catalytic methods for their generation and their high reactivity once formed. Recently, the first asymmetric version was achieved by Nelson et al.[54], which was enabled by the imidodiphosphorimidate (IDPi) organocatalyst[55] and led to the enantioselective synthesis of [4.3.0] and [3.2.1] bicycles. To our best knowledge, metal-catalyzed asymmetric C(sp³)−H insertion via vinyl cations has not been explored yet. Inspired by our recent studies on the chiral copper(I)-catalyzed diyne cyclization via vinyl cations[56–62] and the application of ynamides in N-heterocycle synthesis[63–70], we envisioned that copper-catalyzed cyclization of alkyl-substituted N-propargyl ynamides might generate the highly reactive vinyl cation intermediates, which could serve as the equivalent of donor-donor carbenes, and undergo an intramolecular [1,5]-H migration, eventually leading to the enantioenriched pyrrole derivatives and diynes through kinetic resolution. Here we describe the realization of such a chiral copper-catalyzed tandem diyne cyclization/unactivated C(sp³)−H insertion reaction (Fig. 1c). The reaction undergoes a kinetic resolution of the prochiral N-propargyl ynamides, and allows the efficient and practical synthesis of a range of polycyclic pyrroles in good to excellent yields with generally excellent enantioselectivities. Meanwhile, the remaining diyne substrates can be recovered with generally good to excellent yields and enantioselectivities. Significantly, this protocol not only demonstrates a kinetic resolution reaction based on diyne cyclization but also constitutes a metal-catalyzed enantioselective unactivated C(sp³)−H functionalization via vinyl cation. Theoretical calculations further support the reaction mechanism and the origin of enantioselectivity.

## Results

To prohibit the background aromatic C−H insertion reaction[56], 2,6-dimethylphenyl-substituted N-propargyl ynamide **1a** was chosen as the model substrate under our previous related reaction conditions[56–62], and selected results are listed in Table 1. First, some typical chiral biphosphine ligands and bisoxazoline (BOX) ligands were employed as chiral ligands in the presence of 10 mol % of Cu(MeCN)₄PF₆ as the catalyst and 12 mol % of NaBArᶠ₄ as the additive. To our delight, the use of various chiral ligands resulted in the kinetic resolution of ynamide **1a** (Table 1, entries 1–6), and the use of BOX ligand **L6** afforded the desired chiral product **2a** with 88% ee (Table 1, entry 6). To further improve the enantioselectivity of this reaction, BOX ligands bearing different dibenzyl groups as the side-arm were investigated. These side-arm BOX (SaBOX) ligands[71] significantly impacted the reactivity and enantioselectivity of the reaction. Screening of the SaBOX ligands **L7**–**L10** (Table 1, entries 7–10) showed that the use of **L10** led to the formation of **2a** in 94% ee and 57.5 selectivity (s) factor (Table 1, entry 10). Typical solvents such as THF, toluene, PhCl, and ᵐxylene were also screened (Table 1, entries 11–14). It was found that using ᵐxylene as the solvent further increased the enantioselectivity of **2a** to 95%, and the selectivity factor could be improved to 77.6 (Table 1, entry 14). Gratifyingly, an apparent temperature influence was observed. Decreasing the reaction temperature to 0 °C allowed the formation of the desired **2a** in 95% ee with 145.7 selectivity factor, and the diyne substrate **1a** was also recovered in 95% ee (Table 1, entry 15).

With the optimal reaction conditions in hand (Table 1, entry 15), we started to investigate the scope of this tandem diyne cyclization/C(sp³)−H insertion via kinetic resolution under chiral copper catalysis. As depicted in Fig. 2, the reaction proceeded efficiently with a wide range of N-propargyl ynamides **1**. First, ynamides bearing different N-protecting groups, such as Ts, Mbs, SO₂Ph, Bs, and Ms, were well tolerated under the optimized conditions, affording the chiral polycyclic pyrroles (−)-**2a**–(−)-**2e** and recovered ynamides (+)-**1a**–(+)-**1e** in good yields with good to excellent enantioselectivities. Next, we turned our attention to the aryl groups at the propargylamide. Substrates with

**Table 1 | Optimization of reaction conditions for tandem diyne cyclization/C(sp³)–H insertion of N-propargyl ynamide 1a**

| Entry | L | Reaction conditions | Ee$_s$[a] | Ee$_p$[a] | C (%)[b] | s[c] |
|---|---|---|---|---|---|---|
| 1 | L1 | DCM, 10 °C, 1h | 43 | 71 | 37.7 | 8.9 |
| 2 | L2 | DCM, 10 °C, 1h | 91 | 44 | 67.4 | 7.4 |
| 3 | L3 | DCM, 10 °C, 1h | 21 | 10 | 67.7 | 1.5 |
| 4 | L4 | DCM, 10 °C, 1h | 15 | 75 | 16.7 | 8.1 |
| 5 | L5 | DCM, 10 °C, 1h | <1 | <1 | / | / |
| 6 | L6 | DCM, 10 °C, 1h | 19 | 88 | 17.8 | 18.9 |
| 7 | L7 | DCM, 10 °C, 3 h | 24 | 82 | 22.6 | 12.8 |
| 8 | L8 | DCM, 10 °C, 3 h | 18 | 95 | 15.9 | 46.5 |
| 9 | L9 | DCM, 10 °C, 3 h | 12 | 92 | 11.5 | 27.0 |
| 10 | L10 | DCM, 10 °C, 3 h | 57 | 94 | 37.7 | 57.5 |
| 11 | L10 | THF, 10 °C, 3 h | 98 | 70 | 58.3 | 24.9 |
| 12 | L10 | toluene, 10 °C, 3 h | 57 | 85 | 40.1 | 21.9 |
| 13 | L10 | PhCl, 10 °C, 3 h | 34 | 93 | 26.8 | 38.4 |
| 14 | L10 | ᵐxylene, 10 °C, 3 h | 66 | 95 | 41.0 | 77.6 |
| **15** | **L10** | **ᵐxylene, 0 °C, 14 h** | **95** | **95** | **50.0** | **145.7** |

Reaction conditions: **1a** (0.05 mmol), Cu(MeCN)₄PF₆ (0.005 mmol), **L** (0.006 mmol), NaBArF₄ (0.006 mmol), solvent (1 ml), in Schlenk tubes.
*Ts* p-toluenesulfonyl, NaBArF₄ sodium tetrakis[3,5-bis(trifluoromethyl)phenyl]borate.
[a]Ees are determined by HPLC analysis.
[b]Conversion (C) = ee$_s$/(ee$_s$ + ee$_p$).
[c]Selectivity factor (s) = ln[1 − C(1 + ee$_p$)]/ln[1 − C(1 − ee$_p$)].
Bold values indicate the optimal conditions.

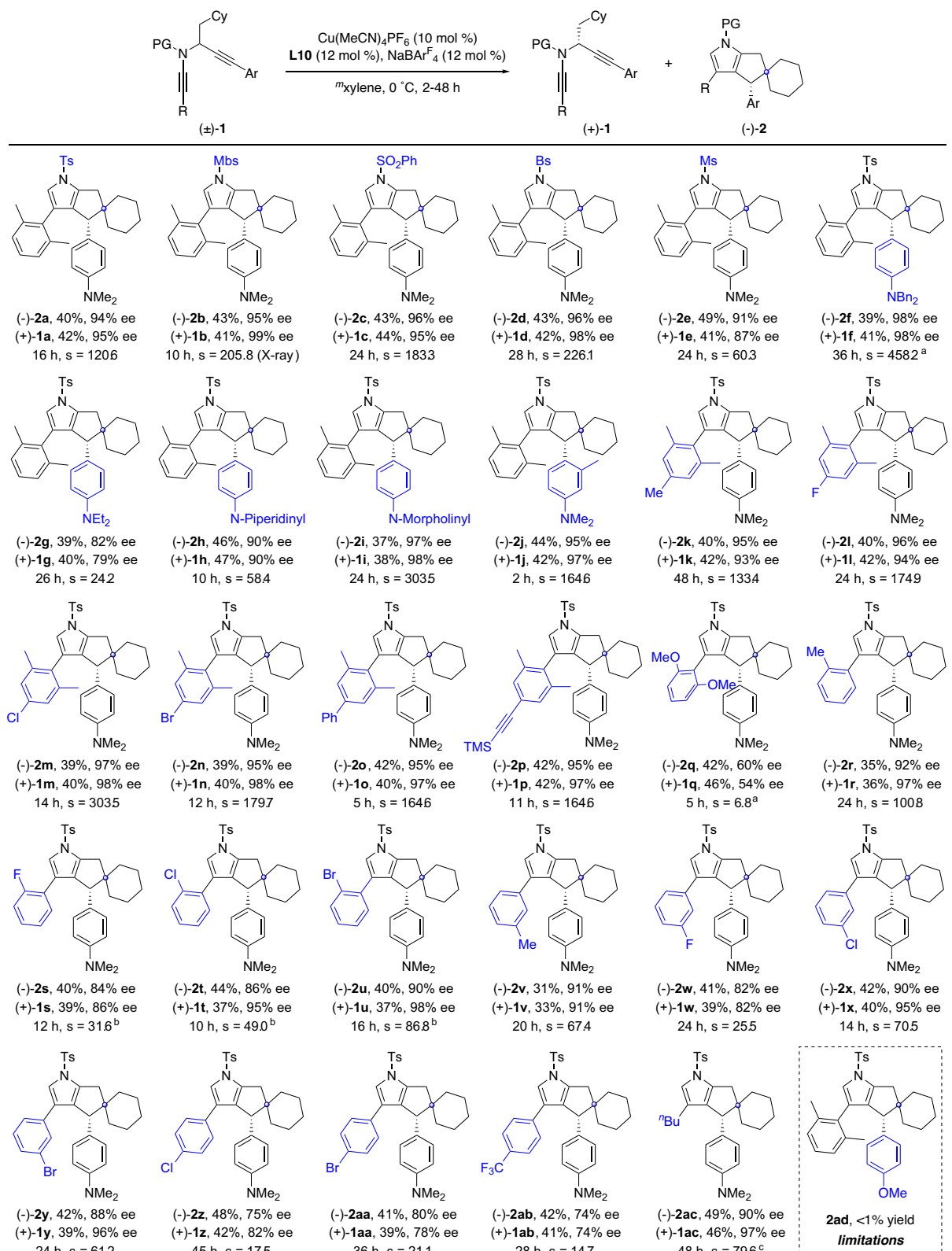

**Fig. 2 | Scope of tandem diyne cyclization/C(sp³)–H insertion of *N*-propargyl ynamides 1.** Reaction conditions: **1** (0.2 mmol), Cu(MeCN)₄PF₆ (0.02 mmol), NaBArᶠ₄ (0.024 mmol), **L10** (0.024 mmol), *m*xylene (4 ml), 0 °C, 2–48 h, in Schlenk tubes; yields are those for the isolated products; ees are determined by HPLC analysis. [a]30 °C. [b]**L8** instead of **L10**. [c]15 °C. Mbs = 4-methoxybenzenesulfonyl, Bs = 4-bromobenzenesulfonyl, PMB = 4-methoxyphenyl.

various heteroatom-containing electron-rich aryl groups, including Bn₂N-, Et₂N-, *N*-piperidinyl, and *N*-morpholinyl, were converted into the expected polycyclic pyrroles (−)-**2f**–(−)-**2i** in good yields with generally good to excellent enantioselectivities. Notably, the Me group at the *ortho*-position of the aryl ring had no impact on the enantioselectivity of the reaction, affording the recovered ynamide (+)-**1j** in 42% yield with 97% ee, and the polycyclic pyrrole (−)-**2j** in 44% yield with 95% ee. Then, a series of substituted 2,6-dimethylphenyl groups at the

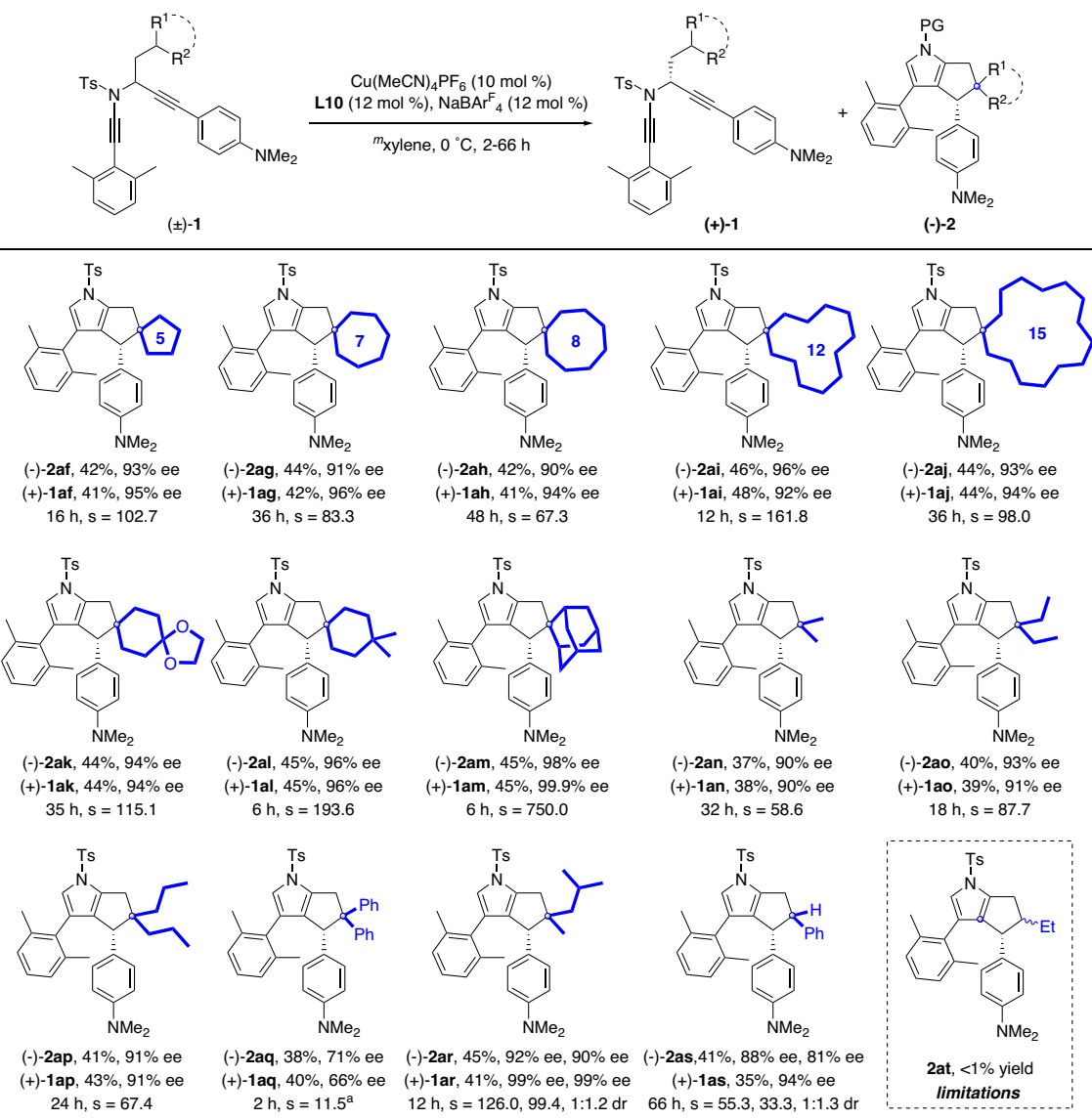

**Fig. 3 | Scope of tandem diyne cyclization/C(sp³)−H insertion of *N*-propargyl ynamides 1.** 1 (0.2 mmol), Cu(MeCN)₄PF₆ (0.02 mmol), NaBArᶠ₄ (0.024 mmol), **L10** (0.024 mmol), *m*xylene (4 ml), 0 °C, 2–66 h, in Schlenk tubes; yields are those for the isolated products; ees are determined by HPLC analysis; drs are determined by crude ¹H-NMR. ª−20 °C.

ynamide moiety were explored. Diynes bearing both electron-withdrawing and -donating groups on the aromatic ring, such as Me, F, Cl, Br, Ph, and alkynyl, were suitable substrates in this tandem reaction, furnishing the desired (−)-**2k**−(−)-**2p** in 39–42% yields with 95–97% ees, and chiral substrates (+)-**1k**−(+)-**1p** could also be recovered in 40–42% yields with 93–98% ees. 2,6-Dimethoxy-substituted substrate was also attempted, and the desired product (−)-**2q** could be obtained. However, the enantiocontrol was not satisfying. To our surprise, mono-substituted substrates, such as *ortho* methyl-, fluoro-, chloro-, and bromo-substituted diynes, could also deliver the target products (−)-**2r**−(−)-**2u** in good yields and enantioselectivities. Furthermore, the reaction also occurred smoothly with *meta*-and *para*-substituted substrates, and the corresponding sp³ C−H insertion products (−)-**2v**−(−)-**2ab** were obtained in moderate to good yields but with slightly decreased enantioselectivities. It is worth noting that these mono-substituted diynes failed to furnish the desired products in our previous diyne cyclization[60] probably due to the competing aromatic C(sp²)−H insertion[56]. However, in our present mono-substituted cases, the corresponding C(sp²)−H insertion byproducts were almost not observed. Finally, it was found that the alkyl-

substituted diyne substrate could also be converted into the desired product (−)-**2ac** in 49% yield and 90% ee, and (+)-**1ac** could be recovered in 46% yield and 97% ee. Our attempts to extend the reaction to the PMP- and TIPS-tethered substrates (**2ad**, **2ae**) resulted in the formation of complicated mixtures. The molecular structure of **2b** was confirmed by X-ray diffraction (Supplementary Fig. 9).

Having established the reactivity of various substituted diyne moieties, we sought to investigate the reactivity with diynes bearing different C(sp³)−H bonds. As shown in Fig. 3, different ring sizes were first examined, including 5-, 7-, 8-, 12-, and 15-membered rings. These diynes underwent smooth tandem cyclization/C(sp³)−H insertion to furnish the desired enantioenriched products (−)-**2af**−(−)-**2aj** in 42–46% yields and 90–96% ees, and the corresponding chiral substrates (+)-**1af**−(+)-**1aj** were also recovered in good yields and good to excellent enantioselectivities. Next, diynes with different 6-membered rings were investigated, and the target products ((−)-**2ak**, (−)-**2al**) were obtained in good to excellent yields and enantioselectivities. Particularly, adamantane-substituted substrate was also applicable to this reaction, furnishing the corresponding polycyclic pyrrole (−)-**2am** in 45% yield and 98% ee with selectivity factor up to 750.0. In addition to

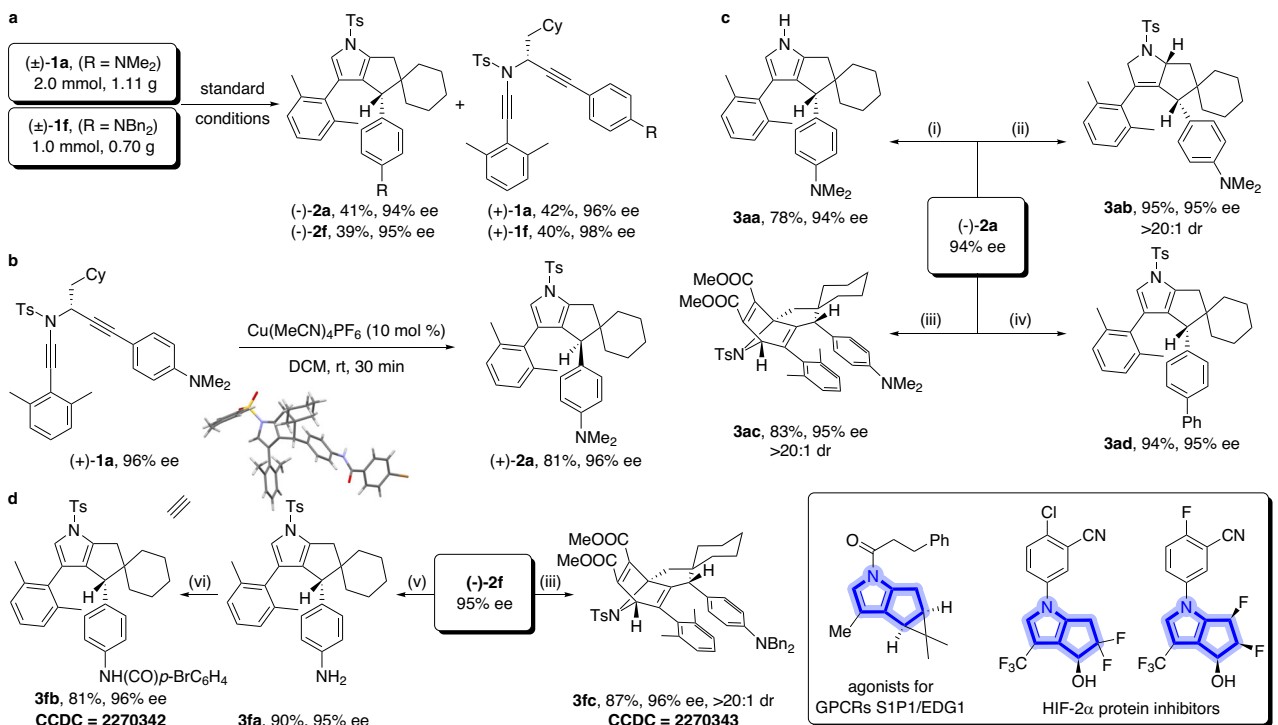

**Fig. 4 | Scale-up reaction and product elaborations. a** Preparative-scale synthesis of products (−)-**2a** and (−)-**2f**. **b** Synthesis of the chiral product (+)-**2a** from chiral diyne (+)-**1a**. **c** Conversion of product (−)-**2a** into compounds **3aa**–**3ad**. **d** Conversion of product (−)-**2f** into compounds **3fa**–**3fc**. Reagents and conditions: (i) KOH (5 equiv), THF/MeOH = 1:1, 80 °C, 8 h. (ii) NaBH₃CN (5 equiv), DCM/TFA =

10:1, 0 °C, 1 h. (iii) dimethyl acetylenedicarboxylate (10 equiv), toluene, 100 °C, 12 h. (iv) PhMgBr (1.2 equiv), Pd(PPh₃)₂Cl₂ (5 mol %), THF, rt, 30 min. (v) Pd/C (10% w/w), H₂ (1 atm), EtOH/EtOAc = 1:1, 80 °C, 5 h. (vi) *p*-BrC₆H₄COCl (1.5 equiv), Et₃N (2 equiv), DCM, rt, 2 h.

the C(sp³)–H bonds of cyclic substrates, we also screened diynes containing the general C(sp³)–H bond of a tertiary carbon center, e.g., dimethyl, diethyl, and dipropyl substituted substrates. To our delight, the expected chiral products (−)-**2an**–(−)-**2ap** were delivered in 37–41% yields and 90–93% ees. Then, the diphenyl-substituted substrate with high reactivity was tried, and the target product (−)-**2aq** was formed in good yield but with significantly decreased enantioselectivity (71% ee). Moreover, an initial attempt to extend the reaction to substrate with an asymmetric 3 °C(sp³)–H bond resulted in the formation of the desired product (−)-**2ar** in good yield and enantioselectivity but only with poor diastereoselectivity (1:1.2). Besides the C(sp³)–H bond of a tertiary carbon center, the C(sp³)–H bond on a secondary carbon center was also studied. Interestingly, this substrate also afforded the target chiral product (−)-**2as** and recovered substrate (+)-**1as** in good yields and enantioselectivities but with only 1:1.3 diastereoselectivity. Furthermore, an unactivated secondary C(sp³)–H bond was also attempted, but failed to obtain the target product **2at**. A likely reason for this is that the secondary C(sp³)–H bonds are electronically unfavoured compared with the tertiary C(sp³)–H bonds.

Preparative-scale synthesis and the synthetic utility of this tandem diyne cyclization/C(sp³)–H insertion were then explored. As shown in Fig. 4, the kinetic resolution of the model substrate **1a** (2.0 mmol, 1.11 g) under the standard conditions afforded the desired product (−)-**2a** in 41% yield with 94% ee, and (+)-**1a** was recovered in 42% yield with 96% ee (Fig. 4a). Meanwhile, **1f** (1.0 mmol, 0.70 g) could undergo this kinetic resolution to deliver the target product (−)-**2f** in 39% yield with 95% ee and the recovered (+)-**1f** in 40% yield with 98% ee (Fig. 4a). Of note, the chiral diyne (+)-**1a** could be readily converted into the chiral product (+)-**2a** in 81% yield without employing the SaBOX ligand, and almost no erosion of the enantiopurity of the product was observed (Fig. 4b). Thus, both enantiomers of polycyclic pyrroles **2** could be synthesized by this strategy. Then, further transformations of

the chiral products, such as (−)-**2a** and (−)-**2f**, were also conducted. First, the Ts group of (−)-**2a** was easily removed under basic conditions, affording the expected pyrrole product **3aa** in 78% yield (Fig. 4c, (i)). Subsequently, the treatment of (−)-**2a** with NaBH₃CN as the reductant led to the desired dihydropyrrole **3ab** bearing two stereocenters in 95% yield with >20:1 dr (Fig. 4c, (ii)). Interestingly, a bridged polycyclic skeleton **3ac** containing three stereocenters was also furnished in 83% yield with > 20:1 dr through a Diels-Alder (DA) reaction of the pyrrole ring of (−)-**2a** with dimethyl acetylenedicarboxylate (Fig. 4c, (iii)). In addition, the NMe₂ group of pyrrole (−)-**2a** could be readily converted into the aryl group in 94% yield via a Pd-catalyzed cross-coupling with the aryl Grignard reagent (Fig. 4c, (iv)). Furthermore, the NBn₂ group of pyrrole (−)-**2f** could also be transformed into the free aniline product **3fa** through a debenzylation process in 90% yield (Fig. 4d, (v)), which could undergo further condensation with the corresponding acyl chloride to deliver the amide **3fb** in 81% yield (Fig. 4d, (vi)). Similarly, the treatment of (−)-**2f** with dimethyl acetylenedicarboxylate allowed the formation of the expected **3fc** in 87% yield (Fig. 4d, (iii)). Importantly, almost no erosion of the enantiopurity of these products was observed, and excellent diastereoselectivities (dr > 20:1) were achieved in all these cases. Notably, these bicyclo pyrrole skeletons can be found in a variety of bioactive molecules and may have some medicinal interest[72,73]. The absolute configuration of **3fb** and the relative configuration of **3fc** were confirmed by X-ray diffraction (Supplementary Figs. 10 and 11), and the former also determined the absolute configuration of the chiral polycyclic pyrroles **2**.

To probe the reaction mechanism, several control experiments were next carried out. We performed a deuterium labeling experiment with the substrate (±)-[D]*-**1a**, and found that the deuterium atom was utterly retained in (−)-[D]*-**2a** under the standard conditions (Fig. 5a). Subsequently, the treatment of diyne (±)-**1a** with 10 equiv of D₂O under the standard conditions revealed a significant deuterium

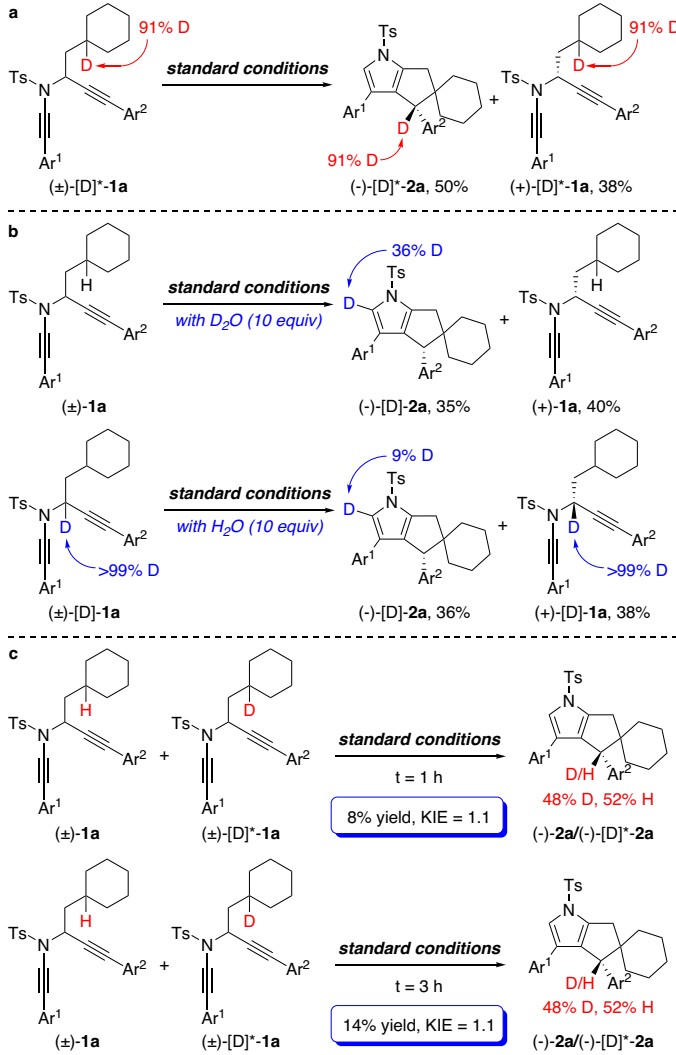

**Fig. 5 | Control experiments. a** The reaction of (±)-[D]*-**1a** under the standard conditions. **b** Hydrogen/deuterium exchange experiments of (±)-[D]-**1a** and (±)-**1a**. **c** KIE experiments. Ar[1], 2,6-dimethylphenyl; Ar[2], 4-NMe₂phenyl.

H-abstraction transition state with large barrier height (subsequent H-migration and demetallation step is omitted for slight barrier height). Afterwards, the recoordination of the desorbed catalyst Cu**L10** to diyne (+)-**1a** occurs and leads to the product (+)-**2a**, with much less reactivity due to a much larger RDS barrier height (28.2 kcal/mol) compared to the RDS barrier height leading to (−)-**2a** (24.9 kcal/mol), thus accounting for the kinetic behaviors of the reaction. The calculated energy profiles of the reaction with both enantiomers of **1a** indicate that the irreversible enantiodetermining step is the intramolecular cyclization step. The optimized structures and barrier height of the enantioselective intramolecular cyclization transition states (−)-**TS_A** and (+)-**TS_A** are also exhibited in Fig. 6. Transition state (+)-**TS_A** (from (+)-**1a**) is disfavored because of the H-H steric repulsion between the Ts group and the side arm of SaBOX ligand as a result of the more proximal orientation of the bulky Ts group. Such steric repulsions are not present in the favored transition state (−)-**TS_A** (from (−)-**1a**), which leads to the difference in free energy between the two competing transition states.

## Discussion

In summary, we have disclosed a chiral copper-catalyzed tandem diyne cyclization/C(sp³)–H insertion reaction via kinetic resolution, furnishing chiral polycyclic pyrroles and diynes in generally excellent enantioselectivities. Importantly, this reaction not only represents a kinetic resolution reaction based on diyne cyclization and a metal-catalyzed enantioselective unactivated C(sp³)−H functionalization via vinyl cation, but also constitutes an asymmetric unactivated C(sp³)−H insertion based on alkynes. In addition, theoretical calculations have been carried out to understand the reaction mechanism and the origin of enantioselectivity. We believe these findings will offer further perspectives and explorations in the field of kinetic resolution and enantioselective C(sp³)−H functionalization based on alkynes and vinyl cations.

## Methods

### General

For ¹H, ¹³C, and ¹⁹F nuclear magnetic resonance (NMR) spectra of compounds in this manuscript and details of the synthetic procedures as well as more reaction condition screening, see Supplementary Information.

### General procedure for the chiral copper-catalyzed kinetic resolution of diynes 1

The powered Cu(MeCN)₄PF₆ (0.02 mmol, 7.5 mg), **L10** (0.024 mmol, 17.4 mg), and NaBAr^F₄ (0.024 mmol, 21.3 mg) were introduced into an oven-dried Schlenk tube under argon atmosphere. After ᵐxylene (2 ml) was injected into the Schlenk tube, the solution was stirred at rt under the argon atmosphere for 2 h. Then the reaction was cooled to 0 °C, and *N*-propargyl ynamide **1** (0.2 mmol) in ᵐxylene (2 ml) was introduced into the system dropwise. The resulting mixture was stirred at indicating temperature and the progress of the reaction was monitored by TLC or HPLC. After concentration in vacuo, the residue was purified by flash chromatography on silica gel (eluent: hexanes/EA or hexanes/DCM) to give the final product (−)-**2** and (+)-**1**.

### Data availability

Data for the crystal structures reported in this paper have been deposited at the Cambridge Crystallographic Data Centre (CCDC) under the deposition numbers CCDC 2270341 (**2b**), 2270342 (**3fb**) and 2270343 (**3fc**). Copies of these data can be obtained free of charge via www.ccdc.cam.ac.uk/data_request/cif. All other data supporting the findings of this study, including experimental procedures and compound characterization, are available within the paper and its Supplementary Information files or from the corresponding authors on request. Source data are provided with this paper.

incorporation into the pyrrole ring of (−)-[D]-**2a** (Fig. 5b). This observation was also found in the reaction of (±)-[D]-**1a** with 10 equiv of H₂O as additive under the standard conditions (Fig. 5b). These results are consistent with our previous work[58], in which water-assisted proton transfer was presumably involved in the formation of pyrrole moiety. Finally, the kinetic isotope effect (KIE) experiment was conducted with the mixture of diynes (±)-**1a** and (±)-[D]*-**1a**, and the result ($k_H/k_D = 1.1$) suggests that the cleavage of C(sp³)−H bond is not the rate-determining step (Fig. 5c).

Based on the above experimental results, density functional theory (DFT) calculations have been performed to gain more insights into the mechanism for the enantioselective synthesis of product **2a** from **1a** and elucidate the kinetic behavior of the reaction and the origin of enantioselectivity. As is exhibited in Fig. 6, similar to our previously reported studies[56–62], at the beginning of the reaction the coordination of Cu**L10** copper species preferentially to the diyne (−)-**1a** leads to a much more stable [Cu]-bound intermediate (−)-**A**, which subsequently triggers an intramolecular cyclization to afford the vinyl cation intermediate (−)-**B** via a transition state (−)-**TS_A**. Then, a C(sp³)−H insertion process gives the copper carbene intermediate (−)-**C** via a transition state (−)-**TS_B**. Finally, as reported in our previous publications[56–62], a H₂O-assisted [1,4]-H migration and demetallation process occurs to afford the product (−)-**2a**, via a transition state (−)-**TS_C** referring to the

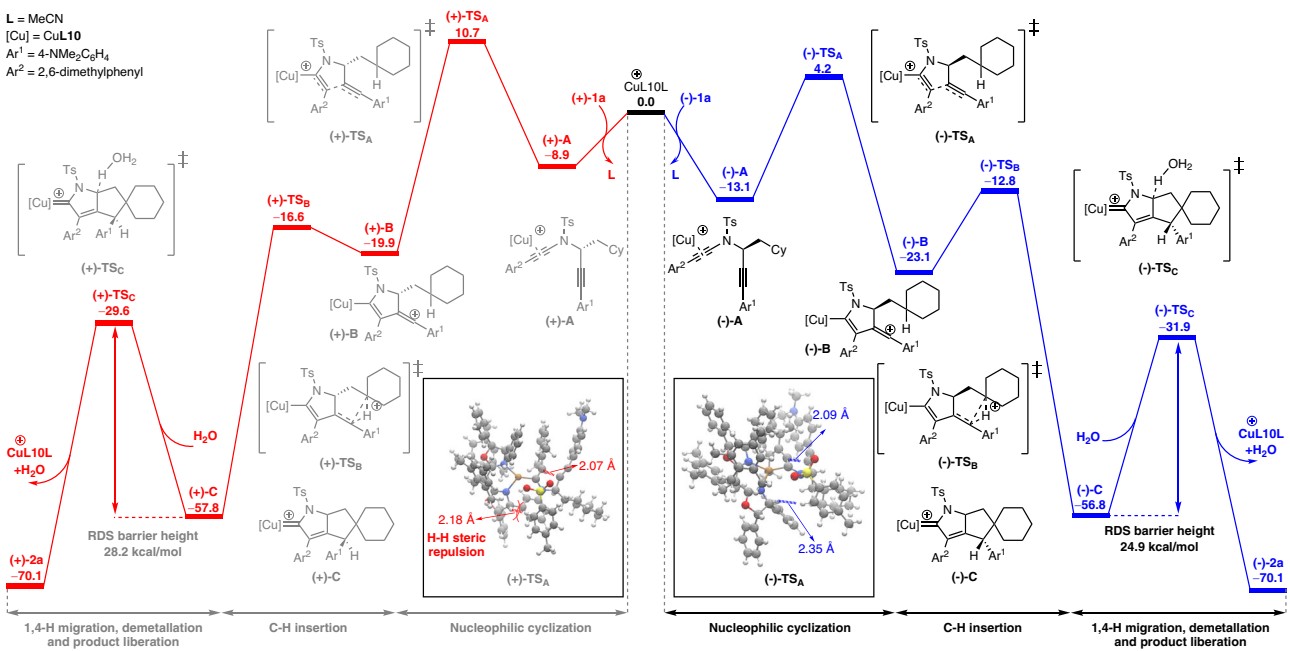

**Fig. 6 | Plausible reaction mechanism.** Relative free energies (ΔG, in kcal/mol) of all the transition states and intermediates were computed at the SMD(solvent = $^m$xylene)-PBE0-D3/Def2-TZVP//SMD(solvent = $^m$xylene)-B3LYP-D3/6-31G(d,p) level of theory. Color code: red = O; white = H; gray = C; yellow = S; blue = N; brown = Cu.

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

## Acknowledgements

We are grateful for financial support from the National Natural Science Foundation of China (22125108, 22331004, 22121001 and 92056104), the President Research Funds from Xiamen University (20720210002), and NFFTBS (J1310024).

## Author contributions

Y.-B.C., Z.-Q.W., R.C., and B.Z. performed experiments. L.-G.L. and X.L. performed DFT calculations. L.-W.Y. conceived and directed the project and wrote the paper. All authors discussed the results and commented on the manuscript.

## Competing interests

The authors declare no competing interests.
