## [Peer Review File · Nature Communications]

Enantioselective functionalization of unactivated C(sp³)–H bonds through copper-catalyzed diyne cyclization by kinetic resolutionREVIEWER COMMENTS

Reviewer #1 (Remarks to the Author):

The work by Ye and coworkers demonstrates a chiral copper-catalyzed tandem diyne cyclization followed by C(sp³)-H insertion reaction via kinetic resolution. The synthetic method allows making a chiral polycyclic pyrrole and diyne derivatives. Notably, the reaction condition ended up with excellent enantioselectivities and selectivity factors. Yields are good to excellent. A plausible mechanism has been shown which is further supported by control experiments and DFT studies.

The development of synthetic method that involves remote C(sp³)-H insertion is always challenging. Moreover, the approach that does not need a directing group in the presence of high valent metal species in the slippage of C(sp³)-H is always noteworthy. In this work, a simple cationic Cu(I) along with chiral box ligand could able to do enantioselective intramolecular cyclization followed by vinyl cation driven inert C(sp³)-H insertion.

The synthetic method ended up with the formation of spiro system through kinetic resolution of the suitable enantiomers is interesting. The enantiodetermining step is the intramolecular cyclization. Despite having the exocyclic vinyl cation, the chiral copper pocket could able to retain the selectivity till the C(sp³)-H insertion step, which is notable.

The concept of copper catalyzed intramolecular cyclization of yne-tethered ynamide through the generation of vinyl cation is well established by Long-Wu Ye group. They carry forwarding their developed methodology and this manuscript showcases an impressive legacy in continuing their work in the field of diyne cyclization.

This work is suitable for the publication in Nature Communications. Some queries need attention.

→ It is too contradictory to believe that propargyl terminus having para methoxy group attached with alkyne is giving <1% of the desired pyrrole derivative 2ac. All the substrates are made by taking starting diyne having -NMe₂ group at the propargyl terminal. This substrate specific dependency should be clarified. A DFT energy calculation with -OMe substituent should be perfect to explain this limitation.

→ One of the X-Ray structures should be included inside the manuscript rather than putting all of them in the supporting information file.

→ A few attempts with the diyne having ortho substituents other than methyl group in the ynamide terminus should be checked and the enantiomeric outcomes should be mentioned.

→ A suitable reason for the failure to get 2as should be addressed in the manuscript body.

→ For Figure 4; the manuscript body write up should be more specific. As for example, Figure 4c contains four different reaction conditions (i, ii, iii, iv), but for all the step it is written Figure 4c in the manuscript body. It should be written as Figure 4c, (i) and so on.

→ A due citation to the kinetic resolution of C-H bond [Chem. Sci. 2021, 12, 14863. The authors should cite some of the original work in the cyclization modules of yne-tethered ynamides [Chemistry-A

European Journal 2013, 19, 9428-9433; Organic letters 2015, 5662-5665; Organic Letters 2022, 24, 8289-8294; Angewandte Chemie International Edition 2019, 58, 2365-2370; Angewandte Chemie International Edition 2019, 58, 2289-2294]

Reviewer #2 (Remarks to the Author):

This manuscript written by Ye and co-workers reports a kinetic resolution of diynes via enantioselective C-H insertion using a chiral copper catalyst. Recently, vinyl cation formation has attracted much attention, and some interesting cascade reactions of diynes and related alkynes based on vinyl cation formation have been reported. However, the catalytic enantioselective C(sp³)-H functionalization through vinyl cations is unprecedented. Furthermore, the kinetic resolution of alkynes via direct diyne cyclization has not yet been realized.

In this study, the authors investigated a copper-catalyzed kinetic resolution of racemic N-propargyl ynamides and found that the bisoxazoline ligand L10-based copper complex promotes the desired kinetic resolution to give the chiral polycyclic pyrroles and the recovered ynamides in mostly >60% yields. The reaction scope has been well examined using the substrates with various substitution patterns. It should be noted that the desired reaction proceeded efficiently even with the substrates that can induce arylation of vinyl cations. The proposed reaction mechanism is well supported by deuterium experiments and DFT calculations.

This study first demonstrated enantioselective C-H activation reaction via vinyl cation. Thus, this well-written paper is suitable for publication in Nature Communication after some minor revisions noted below:

- TOC graphic and Fig. 1c: the structure of the propargyl ynamides should be depicted in the same orientation as other reaction schemes.
- Please comment on the reaction using a phenyl-substituted propargyl ynamide (R = Ph).
- Fig. 4: CCDC number for 3fc should be added (similarly to 3fb).
- Fig. 6 is not intuitive because some structures are far from the appropriate positions. For example, the structure of TSA should be placed close to the bar showing the TSA energy values. Intuitive clarity is even more important than reducing dead space.

Reviewer #3 (Remarks to the Author):

This paper discusses the kinetic resolution (KR) of chiral propargylic ynamides producing polycyclic pyrroles in high ee's (Figures 2 and 3). Yields are typically around 40%. This is ok for KR but it shows the inherent lack of practicality compared to dynamic kinetic resolutions that can theoretically achieve 100% yield. A KR protocol is generally not desirable and more of a last resort if everything else fails.

Alternatively, the enantiomers of the chiral starting materials can apparently be resolved and they undergo much faster cyclization to the enantiopure pyrrole products in the presence of an achiral Cu

catalyst (Figure 4b). The KR protocol doesn't really offer much of an advantage. Mechanistically, this work builds on previously reported copper catalyzed asymmetric ring closures and annulations of propargylic ynamides developed by the same group (references 51-57). Some of these reactions proceed via vinyl cation intermediates and have strikingly similar mechanisms. Upscaling is possible and products can be derivatized but it is not clear how this chemistry would provide better, if any, synthetic access to the medicinally relevant structures shown in Figure 4. The authors were successful with making compound 2a (Figure 2) but not 2ac (Figure S1). This is surprising and requires an explanation. Based on the SI the synthesis of the propargylic ynamide starting materials from potentially commercially available alcohols requires six reactions and four chromatographic work-ups. This is not attractive. Overall, the work is well executed but unlikely to receive much attention outside of the ynamide community.

REVIEWER COMMENTS

Reviewer #1 (Remarks to the Author):

The work by Ye and coworkers demonstrates a chiral copper-catalyzed tandem diyne cyclization followed by C(sp³)-H insertion reaction via kinetic resolution. The synthetic method allows making a chiral polycyclic pyrrole and diyne derivatives. Notably, the reaction condition ended up with excellent enantioselectivities and selectivity factors. Yields are good to excellent. A plausible mechanism has been shown which is further supported by control experiments and DFT studies.

The development of synthetic method that involves remote C(sp³)-H insertion is always challenging. Moreover, the approach that does not need a directing group in the presence of high valent metal species in the slippage of C(sp³)-H is always noteworthy. In this work, a simple cationic Cu(I) along with chiral box ligand could able to do enantioselective intramolecular cyclization followed by vinyl cation driven inert C(sp³)-H insertion.

The synthetic method ended up with the formation of spiro system through kinetic resolution of the suitable enantiomers is interesting. The enantiodetermining step is the intramolecular cyclization. Despite having the exocyclic vinyl cation, the chiral copper pocket could able to retain the selectivity till the C(sp³)-H insertion step, which is notable.

The concept of copper catalyzed intramolecular cyclization of yne-tethard ynamide through the generation of vinyl cation is well established by Long-Wu Ye group. They carry forwarding their developed methodology and this manuscript showcases an impressive legacy in continuing their work in the field of diyne cyclization.

This work is suitable for the publication in Nature Communications. Some queries need attention.

1. Response to comment (reviewer 1):

1) It is too contradictory to believe that propargyl terminus having para methoxy group attached with alkyne is giving <1% of the desired pyrrole derivative **2ac**. All the substrates are made by taking starting diyne having -NMe₂ group at the propargyl terminal. This substrate specific dependency should be clarified. A DFT energy calculation with -OMe substituent should be perfect to explain this limitation.

- We first thank the reviewer very much for the kind comments and suggestions.
- In fact, compared to its selectivity (such as enantioselectivity or regioselectivity), it has been generally believed that it is difficult to accurately explain and predict the yield of a chemical reaction solely by DFT calculations, because it is considered that the selectivity of a reaction is determined by the kinetic factors such as the barrier height while the reaction yield is often affected by other factors such as thermodynamic factors, reaction conditions, etc.

Fortunately, there is a significant difference in the reaction yield between substrates bearing the *para*-methoxy and *para*-NMe₂ groups. Thus, we performed DFT calculations on the rate-determining step of the reaction after replacing the *para*-methoxy group of the substrate into the *para*-NMe₂ group and found that the

barrier height of the rate-determining step increased significantly to 32.4 kcal/mol from 24.9 kcal/mol, indicating the reaction is unable to occur under experimental conditions, which is consistent with the experimentally obtained reaction yield. And the change in barrier height may somehow be due to the variation of the electronic structure of the molecule caused by the substitution of *para*-NMe₂ to *para*-OMe.

2. Response to comment (reviewer 1):

2) One of the X-Ray structures should be included inside the manuscript rather than putting all of them in the supporting information file.

- As suggested, the X-Ray structure of **3fb** has been inserted into the Fig. 4.

3. Response to comment (reviewer 1):

3) A few attempts with the diyne having *ortho* substituents other than methyl group in the ynamide terminus should be checked and the enantiomeric outcomes should be mentioned.

- We had tried to synthesize the substrates with difluoro-, dichloro-, and dibromo-substituents on the *ortho*-position of the phenyl, but failed to obtain the desired substrate. It is probably due to the steric hindrance effect. Meanwhile, we had also tried to synthesize dimethoxy-substituted substrates. However, the enantiocontrol of

this substrate was not satisfying, and the result is given in the new version of the manuscript. We think it should be caused by the low steric hindrance effect compared with the dimethyl-substituted substrate. This phenomenon can also be concluded from the mono-substituted cases that the bromo-substituted substrate has a better enantiocontrol than chloro- and fluoro-substituted substrates.

4. Response to comment (reviewer 1):

4) A suitable reason for the failure to get **2as** should be addressed in the manuscript body.

- As suggested, we have provided an explanation for the failure of substrate **2at** (**2as** in the old version) with an unactivated secondary C(sp³)-H bond. The following comment has been added into the manuscript: A likely reason for this is that the secondary C(sp³)-H bonds are electronically unfavoured compared with the tertiary C(sp³)-H bonds.

5. Response to comment (reviewer 1):

5) For Figure 4; the manuscript body write up should be more specific. As for example, Figure 4c contains four different reaction conditions (i, ii, iii, iv), but for all the step it is written Figure 4c in the manuscript body. It should be written as Figure 4c, (i) and so on.

- As suggested, we have corrected this and added the detailed information into the manuscript.

6. Response to comment (reviewer 1):

6) A due citation to the kinetic resolution of C-H bond [Chem. Sci. 2021, 12, 14863. The authors should cite some of the original work in the cyclization modules of yne-tethered ynamides [Chemistry–A European Journal 2013, 19, 9428-9433; Organic letters 2015, 5662-5665; Organic Letters 2022, 24, 8289-8294; Angewandte Chemie International Edition 2019, 58, 2365-2370; Angewandte Chemie International Edition 2019, 58, 2289-2294]

- As suggested, we have added the above relevant references in the field of diyne cyclization and kinetic resolution.

Reviewer #2 (Remarks to the Author):

This manuscript written by Ye and co-workers reports a kinetic resolution of diynes via enantioselective C-H insertion using a chiral copper catalyst. Recently, vinyl cation formation has attracted much attention, and some interesting cascade reactions of diynes and related alkynes based on vinyl cation formation have been reported. However, the catalytic enantioselective C(sp³)-H functionalization through vinyl cations is unprecedented. Furthermore, the kinetic resolution of alkynes via direct diyne cyclization has not yet been realized.

In this study, the authors investigated a copper-catalyzed kinetic resolution of racemic N-propargyl ynamides and found that the bisoxazoline ligand **L10**-based copper complex promotes the desired kinetic resolution to give the chiral polycyclic pyrroles and the recovered ynamides in mostly >60 s values. The reaction scope has been well examined using the substrates

with various substitution patterns. It should be noted that the desired reaction proceeded efficiently even with the substrates that can induce arylation of vinyl cations. The proposed reaction mechanism is well supported by deuterium experiments and DFT calculations.

This study first demonstrated enantioselective C-H activation reaction via vinyl cation. Thus, this well-written paper is suitable for publication in Nature Communication after some minor revisions noted below:

7. Response to comment (reviewer 2):

7) TOC graphic and Fig. 1c: the structure of the propargyl ynamides should be depicted in the same orientation as other reaction schemes.

- We first acknowledge the kind recommendation and suggestions from reviewer 2.
- As suggested, the structure of the propargyl ynamides in TOC graphic and Fig. 1c has been reorientated.

8. Response to comment (reviewer 2):

8) Please comment on the reaction using a phenyl-substituted propargyl ynamide (R = Ph).

- In our initial attempts at this work, we had tried the phenyl-substituted propargyl ynamide (R = Ph). However, the chemoselectivity for C(sp³)-H and aromatic C-H bond was not satisfying. We could obtain the desired C(sp³)-H insertion product, but the yield was low (28%), and the enantiocontrol was not very good (76% ee). Thus, we tried to use the dimethyl-substituted substrate as the model substrate to inhibit the aromatic C-H insertion, and more importantly, it could promote the enantiocontrol of the reaction.

9. Response to comment (reviewer 2):

9) Fig. 4: CCDC number for **3fc** should be added (similarly to **3fb**).

- As suggested, we have added the CCDC number for **3fc** in Fig. 4d.

10. Response to comment (reviewer 2):

10) Fig. 6 is not intuitive because some structures are far from the appropriate positions. For example, the structure of TSA should be placed close to the bar showing the TSA energy values. Intuitive clarity is even more important than reducing dead space.

- As suggested, we have rearranged Fig. 6 and put the structure as close as possible to the bar showing the energy values.

Reviewer #3 (Remarks to the Author):

This paper discusses the kinetic resolution (KR) of chiral propargylic ynamides producing polycyclic pyrroles in high ee's (Figures 2 and 3). Yields are typically around 40%. This is ok for KR's but it shows the inherent lack of practicality compared to dynamic kinetic resolutions that can theoretically achieve 100% yield. A KR protocol is generally not desirable and more of a last resort if everything else fails. Alternatively, the enantiomers of the chiral starting materials can apparently be resolved and they undergo much faster cyclization to the enantiopure pyrrole products in the presence of an achiral Cu catalyst (Figure 4b). The KR protocol doesn't really offer much of an advantage. Mechanistically, this work builds on previously reported copper catalyzed asymmetric ring closures and annulations of propargylic ynamides developed by the same group (references 51-57). Some of these reactions proceed via vinyl cation intermediates and have strikingly similar mechanisms. Upscaling is possible and products can be derivatized but it is not clear how this chemistry would provide better, if any, synthetic access to the medicinally relevant structures shown in Figure 4. The authors were successful with making compound **2a** (Figure 2) but not **2ac** (Figure S1). This is surprising and requires an explanation. Based on the SI the synthesis of the propargylic ynamide starting materials from potentially commercially available alcohols requires six reactions and four chromatographic work-ups. This is not attractive. Overall, the work is well executed but unlikely to receive much attention outside of the ynamide community.

11. Response to comment (reviewer 3):

- First, I really do not agree with the comment of “A KR protocol is generally not desirable and more of a last resort if everything else fails”. Instead, kinetic resolution (KR) is a very important way for the synthesis of chiral compounds. Please see the review papers (refs. 19-22) in the manuscript and also other related review papers (*Eur. J. Org. Chem.* 2012, 1471–1493; *Acc. Chem. Res.* 2016, 49, 2807–2821; *ChemCatChem* 2016, 8, 86–96; *Asian J. Org. Chem.* 2016, 5, 308–320) for details.
- In terms of the comment of “Alternatively, the enantiomers of the chiral starting materials can apparently be resolved and they undergo much faster cyclization to the enantiopure pyrrole products in the presence of an achiral Cu catalyst (Figure 4b). The KR protocol doesn't really offer much of an advantage”, I also do not agree with, as this kind of chiral starting materials can be hardly obtained except by this KR protocol.
- In terms of the comment of “The authors were successful with making compound **2a** (Figure 2) but not **2ac** (Figure S1). This is surprising and requires an explanation”, please see entry 1 of this response letter (also raised by the reviewer 1) for the detailed explanations.
- Importantly, as the above reviewers 1 and 2 mentioned, this protocol not only represents the first example of kinetic resolution reaction based on diyne cyclization but also constitutes the first example of metal-catalyzed enantioselective unactivated C(sp³)-H functionalization via vinyl cation.

REVIEWERS' COMMENTS

Reviewer #1 (Remarks to the Author):

Most of the queries have been addressed by the authors. The DFT calculation linked to the substrate driven transformation is promising. The reaction did not proceed to the expectation for the ortho-substituted compounds; although this is the drawback but this result offers challenges for the further development of the synthetic strategy.

The DFT studies offer valuable insights justifying the reaction failure for para-OMe substituted derivative when compared with the model substrate having para-NMe₂ group. The DFT data should be kept in the supporting information and the explanation details should be highlighted in the manuscript body [After the statement; "Our attempts to extend the reaction to the PMP- and TIPS-tethered substrates (2ad, 2ae) resulted in the formation of complicated mixtures.].

The manuscript is suitable for publication.

Reviewer #2 (Remarks to the Author):

The revised paper adequately addresses all the comments raised by the reviewers 1 and 2. I also agree with the authors' opinion on the reviewer 3's comments.

This paper is now appropriate for acceptance by Nature communications as it stands.